# STEPPING BACK TO SMILES TRANSFORMERS FOR FAST MOLECULAR REPRESENTATION INFERENCE

## ABSTRACT

In the intersection of molecular science and deep learning, tasks like virtual screening have driven the need for a high-throughput molecular representation generator on large chemical databases. However, as SMILES strings are the most common storage format for molecules, using deep graph models to extract molecular feature from raw SMILES data requires an SMILES-to-graph conversion, which significantly decelerates the whole process. Directly deriving molecular representations from SMILES is feasible, yet there exists a performance gap between the existing unpretrained SMILES-based models and graph-based models at large-scale benchmark results, while pretrain models are resource-demanding at training. To address this issue, we propose ST-KD, an end-to-end **SMILES T**ransformer for molecular representation learning boosted by **K**nowledge **D**istillation. In order to conduct knowledge transfer from graph Transformers to ST-KD, we have redesigned the attention layers and introduced a pre-transformation step to tokenize the SMILES strings and inject structure-based positional embeddings. Without expensive pretraining, ST-KD shows competitive results on latest standard molecular datasets PCQM4M-LSC and QM9, with $3\text{-}14\times$ inference speed compared with existing graph models.

## 1 INTRODUCTION

Recent years have witnessed the remarkable progress in the combination of deep learning and molecular science, with regard to tasks including molecular property prediction (Xiong et al., 2019; Li et al., 2021), conformation generation (Xu et al., 2021), and virtual screening for drug discovery (Gentile et al., 2020; Stevenson et al., 2021). Meanwhile, encoding molecules into numerical vectors, or molecular representation, serves as the very first yet important step. As 2D structures of molecules, *i.e.* atom-bond connectivity, can be naturally represented as graphs, neural networks for graph data have become one of the predominant choices. Exploiting network architectures from graph message-passing networks (Gilmer et al., 2017) to Transformers (Rong et al., 2020; Ying et al., 2021), these graph-based methods have reached the state-of-the-art performances on standard benchmark datasets.

However, in real applications such as virtual screening, operations on very large datasets are required, and high-throughput molecular encoders are indispensable. In most existing chemical libraries such as ZINC (Irwin & Shoichet, 2005), molecules are stored with SMILES (Simplified Molecular-Input Line-Entry System, Weininger (1988)), a line notation format with strict generating rules that encodes complete 2D structural information. In order to apply graph-based models to SMILES inputs, an additional SMILES-to-Graph conversion is needed. Operated upon an unparallelizable deterministic algorithm (Weininger, 1990) with high time complexity proved by experiments, this conversion has become a serious efficiency bottleneck that restricts graph-based model from achieving high feature extraction throughput on SMILES datasets. Researchers have also been exploring ways to generate molecular representations (or fingerprints) from SMILES strings in an end-to-end fashion, like the topological ECFP (Rogers & Hahn, 2010) and Transformer-based SMILES language models (Honda et al., 2019; Wang et al., 2019; Chithrananda et al., 2020). However, without pretraining, their overall benchmark results on standard datasets are outperformed by basic graph-based models like GCN (Kipf & Welling, 2016). While for competitive pretrained models like MolFormer (Ross et al., 2021), the inference speed is largely restricted by huge model size, and pretraining requires massive SMILES data and computational resources.

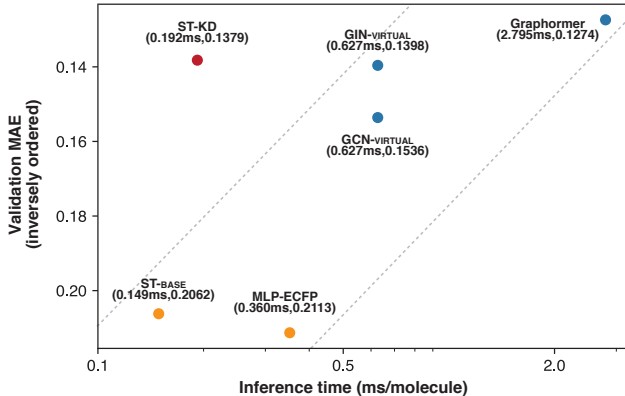

Figure 1: Inference time v.s. accuracy (validation MAE, inversely ordered) on PCQM4M-LSC. ST-KD speeds up 3-14× compared with existing graph-based models with competitive accuracy. Explanations of the dataset and baseline models are in Section 4.

Based on the discussion above, we propose ST-KD, an end-to-end **S**MILES **T**ransformer architecture incorporated with **K**nowledge **D**istillation (KD) from graph Transformers. Considering SMILES implicitly encodes the complete 2D molecular structure, we believe the key reason for the low performances of current SMILES-based models is that, without structure-related supervision, the hidden atom-bond connectivity within SMILES is not well captured in training. During KD from graph Transformer (teacher) to SMILES Transformer (student), the student is forced to mimic the teacher while learning structural knowledge of molecules, for the distilled hidden representations and attention weights are concentrated knowledge the teacher have learned from molecular graphs. This knowledge transfer process is capable of bringing a magnificent performance boost to SMILES Transformer with little loss of efficiency, as shown in Figure 1.

As SMILES is generated from strict grammar rules and contains different types of tokens (like tokens for atoms, bonds, ring-breaks, etc.), applying standard NLP data preprocess methods will lead to faulty tokenization and ambiguous input embeddings. Hence, we build an efficient module to transform SMILES strings into unified token sequences embedded with clear chemical definition, then attach structure-based positional encodings to them. Besides, graph Transformer and SMILES Transformer have different procedures to update token embeddings and compute pairwise attention interactions, so most standard Transformer distillation methods are infeasible. To solve this problem, in addition to feature distillation, we add parameterized attention biases to ST-KD's self-attention layers, and partially supervise on the biases to transfer knowledge in attention blocks. Ablation studies have proved our redesigned distillation techniques are essential to optimize the teaching results.

Directly taking SMILES as input, ST-KD has averted the SMILES-to-graph bottleneck and can run 3-14× faster than graph-based models at inference. Compared to pretrained SMILES models, ST-KD is much more light-weight and efficient at training, without compromising on performance on most tasks. Boosted by the solid network structure and knowledge distillation from cutting-edge graph Transformer (Ying et al., 2021), ST-KD is qualified to deliver competitive, even state-of-the-art results on public molecular benchmarks like MoleculeNet (Wu et al., 2018) and OGB-LSC (Hu et al., 2020). Code for reproducing our results is provided in the supplementary material.

## 2 RELATED WORK

**SMILES and SMILES-based Fingerprint Generators** SMILES (simplified molecular-input line-entry system), first proposed in (Weininger, 1988; Weininger et al., 1989; Weininger, 1990), is a universal line notation format to represent 2D molecular structure. For example, the SMILES string `c1ccccc1C` can specify the structure of toluene. SMILES has complex grammars and all symbols have specific chemical definitions, e.g. `C` and `N` represent carbon atom and nitrogen atom, − and = represent single bond and double bond, `1` and `(` represent (part of) ring-break and branching

bond. An important fact is that the original SMILES system does not create a bijective mapping between the set of all possible SMILES sequences and molecules. A legal SMILES sequence can be translated into one certain molecule, but it is possible for a molecule to have multiple SMILES representations, e.g. both `CCO` and `OCC` specify ethanol.

Before the age of deep learning, researchers have developed multiple SMILES-based molecular fingerprint generators, including the hash-based methods (Glen et al., 2006; Hu et al., 2009; Rogers & Hahn, 2010) and task-driven methods (O'Boyle et al., 2011). Extended-Connectivity Finger-Print (ECFP) (Rogers & Hahn, 2010) is a famous method for efficiently generating topological structure fingerprints for SMILES. On the other side, the recent spurt of deep models has inspired researchers to build SMILES fingerprint generators by training from data without expert knowledge (Xu et al., 2017; Honda et al., 2019; Zhang et al., 2018; Wang et al., 2019; Chithrananda et al., 2020). Treating SMILES as a formal language, most of these methods take NLP models like Transformer (Vaswani et al., 2017) and BERT (Devlin et al., 2018) as backbone, then train the model with pretrain-finetune paradigm for downstream tasks. These methods have shown promising results in specific tasks like molecular property prediction (Ross et al., 2021), chemical reaction prediction (Schwaller et al., 2019) and drug discovery (Honda et al., 2019). As for molecular property prediction, current SMILES-based models can only reach competitive performances against graph-based models with extensive pretraining, like (Ross et al., 2021; Irwin et al., 2021). Compared with large-scale pretraining on massive data, the distillation strategy we employ requires no big data and is much more effective and efficient, which will be discussed in Section 4.2.

**Graph-based Molecular Representation Learning**  Graph-based deep models (Duvenaud et al., 2015; Gilmer et al., 2017; Xiong et al., 2019; Rong et al., 2020; Ying et al., 2021) have taken the leading role in molecular representation learning. Some studies (Cho & Choi, 2019; Klicpera et al., 2020; Song et al., 2020; Li et al., 2021) also begin to enhance molecular representations with 3D conformation. Graphormer (Ying et al., 2021) is a newly proposed graph Transformer model for graph representation learning, which attains the state-of-the-art performance on many public datasets and wins the OGB Large-Scale Challenge Hu et al. (2021) in quantum chemistry regression dataset. In experiments, we pick Graphormer as the teacher model, considering its outstanding performance and Transformer-based architecture.

**Transformer Distillation and Cross-modal Knowledge Transfer**  Transformer (Vaswani et al., 2017) has become one of the most popular building blocks in deep models with its powerful self-attention mechanism, which can capture long-term dependencies within sequences of tokens. Transformer-based pretrain language models like BERT Devlin et al. (2018) exhibit excellent performances but are also computationally expensive. There have been many works that attempt to compress Transformer-based models with knowledge distillation (Jiao et al., 2019; Sun et al., 2019a; 2020; Wang et al., 2020). The distilled knowledge may be soft target probabilities, embedding outputs, hidden representations or attention weight distributions. Existing works have also explored cross-modal knowledge transfer, including contrastive cross-modal distillation (Tian et al., 2019) and graph-to-text distillation (Dong et al., 2020). ST-KD is built upon the Transformer architecture and borrows some insights in distillation methods above. However, since the difference between SMILES and other types of structured knowledge is essential, the distillation techniques we apply are completely redesigned to bridge the knowledge transfer between graph-based models and SMILES-based models.

## 3 METHOD

### 3.1 PRELIMINARIES

**Problem Formulation**  We formulate molecular representation learning as a supervised learning task, which takes notations of molecules as inputs, properties of molecules as supervisions. A molecule $M$ can be denoted as an attributed graph $\mathcal{G}_M = (V, E, n, m, \boldsymbol{X}^V, \boldsymbol{X}^E)$, or a SMILES line notation $\mathcal{S}_M^{\mathrm{Org}} = (s_1, s_2, \ldots, s_l)$, where $V$ is the set of $n$ atoms, $E \subset V \times V$ is the set of $m$ bonds, $\boldsymbol{X}^V = [\boldsymbol{x}_1^V, \ldots, \boldsymbol{x}_n^V]^\top \in \mathbb{R}^{n \times d_V}$ is the atom feature matrix, $\boldsymbol{X}^E = [\boldsymbol{x}_1^E, \ldots, \boldsymbol{x}_m^E]^\top \in \mathbb{R}^{m \times d_E}$ is the bond feature matrix, $s_1, s_2, \ldots, s_l \in \{\texttt{C}, \texttt{N}, \texttt{-}, \texttt{=}, \texttt{1}, \texttt{(}, \ldots\}$ are SMILES symbols.

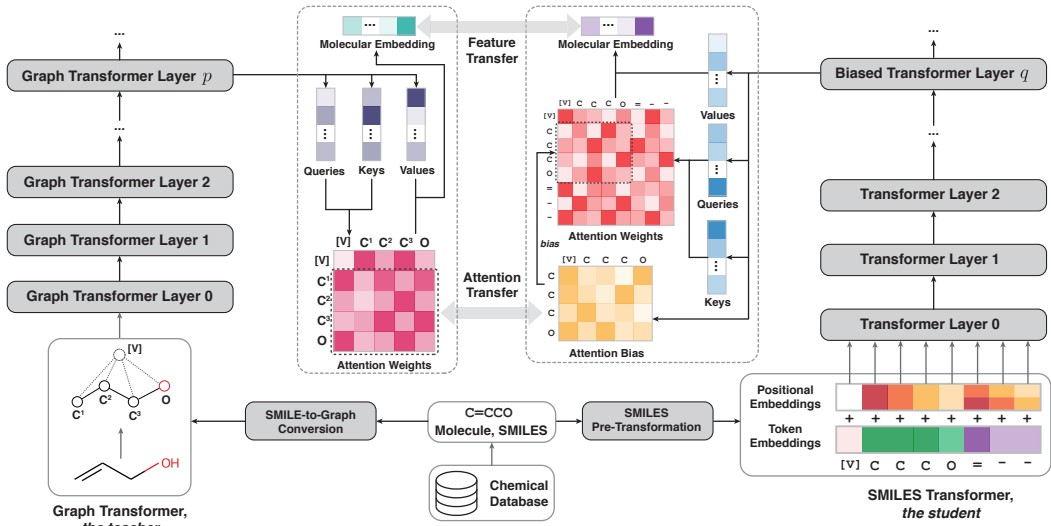

Figure 2: Overview of the proposed ST-KD architecture.

**Overview** Figure 2 is an illustration of the proposed ST-KD architecture. SMILES strings are first transformed to generate initial input embeddings, then sent to the stacked layers. ST-KD is built by standard Transformer layers and biased Transformers layers we introduce in Section 3.3, which are designed to enable the knowledge transfer from graph Transformer attention block to SMILES Transformer.

**Transformer and Multi-head Attention** A Transformer encoder layer is composed of a multi-head attention (MHA) block and feed-forward networks (FFN) with residue connections. Let $\boldsymbol{H} = [\boldsymbol{h}_1, \ldots, \boldsymbol{h}_l]^\top \in \mathbb{R}^{l \times d}$ be the input embeddings, the calculation process of multi-head attention can be formulated as:

$$\boldsymbol{Q}_i = \boldsymbol{H}\boldsymbol{W}_i^Q, \boldsymbol{K}_i = \boldsymbol{H}\boldsymbol{W}_i^K, \boldsymbol{V}_i = \boldsymbol{H}\boldsymbol{W}_i^V, \tag{1}$$

$$\boldsymbol{A}_i = \text{Softmax}(\frac{\boldsymbol{Q}_i\boldsymbol{K}_i^\top}{\sqrt{d_k}}), \text{head}_i = \boldsymbol{A}_i\boldsymbol{V}_i, \text{for } i = 1, \ldots, h, \tag{2}$$

$$\text{MHA}(\boldsymbol{H}) = \text{Concat}(\text{head}_1, \ldots, \text{head}_h)\boldsymbol{W}^O, \tag{3}$$

where $l$ is length of input sequence, $h$ is number of attention heads, $\boldsymbol{W}_i^Q \in \mathbb{R}^{d \times d_k}, \boldsymbol{W}_i^K \in \mathbb{R}^{d \times d_k}, \boldsymbol{W}_i^V \in \mathbb{R}^{d \times d_v}$ and $\boldsymbol{W}^Q \in \mathbb{R}^{hd_v \times d}$ are projection parameter matrices, $d, d_k, d_v$ are the dimension of hidden layers, keys and values.

**Graph Transformer** Graph Transformers (Ying et al., 2021; Dwivedi & Bresson, 2020) generally take graphs as node sequences, update the node representations using attention mechanism and linear layers, and leverage the structural information into the Transformer architecture by positional encodings and attention biases. Given molecule $M$, it should be fed into the graph Transformer model as a featurized graph $(V, E, n, m, \boldsymbol{X}^V, \boldsymbol{X}^E)$ with initial token representations

$$\boldsymbol{H}_0^{\text{Graph}} = [\boldsymbol{x}_0, \boldsymbol{x}_1^V, \ldots, \boldsymbol{x}_n^V]^\top \in \mathbb{R}^{l_{\text{Graph}} \times d_V}, l_{\text{Graph}} = n + 1, \tag{4}$$

where $\boldsymbol{x}_0$ represents for a virtual token for molecule embeddings. Then $\boldsymbol{H}_0^{\text{Graph}}$ is updated by stacked Graph Transformer layers to produce the final result.

### 3.2 SMILES PRE-TRANSFORMATION AND INPUT EMBEDDINGS

In this section, we transform irregular SMILES sequences into a unified representation form, in which all atom and bond tokens (maybe omitted in the original SMILES) are parsed out and embedded with structural information.

**SMILES Pre-Transformation**    At SMILES pre-tranformation, we convert a general SMILES line notation $\mathcal{S}_M^{\text{Org}} = (s_1, s_2, \ldots, s_l)$ into a sequence $\mathcal{S}_M = (s_1^V, \ldots, s_n^V, s_1^E, \ldots, s_m^E)$ that consists of SMILES symbols, where $s_1^V, \ldots, s_n^V \in \{\texttt{C}, \texttt{N}, \texttt{O}, \texttt{c}, \ldots\}$ are $n$ atom symbols, $s_1^E, \ldots, s_m^E \in \{-, =, :, \ldots\}$ are $m$ bond symbols. Atom-bond connectivity is also decoded in this process for computing positional embeddings. We build a SMILES parsing module to perform this step, and we omit its detailed description for simplicity due to the extremely complex SMILES grammar. One can refer to the code for our implementation details.

**Input Representations**    A virtual token $[\texttt{V}]$ is first added to the processed sequence $\mathcal{S}_M$, which represents the molecule embedding and its representation will be updated as normal tokens, like the $[\texttt{CLS}]$ token in BERT. Then for a given token, its input representation is constructed by summing up its corresponding token embedding and positional embedding. We use a SMILES symbol vocabulary to generate learnable token embeddings, and positional embeddings are calculated from the decoded molecular structure in the previous step. We first initialize a learnable parameter tensor $\boldsymbol{E}^{\text{PE}} \in \mathbb{R}^{n_{\max} \times d}$, where $n_{\max}$ is maximum number of atoms of molecules in the dataset, $d$ is the hidden size. Then in the processed SMILES token sequence with special token $([\texttt{V}], s_1^V, \ldots, s_n^V, s_1^E, \ldots, s_m^E)$, for atom token $s_i^V$ with token embedding $\boldsymbol{h}_i^{\text{TE},V} \in \mathbb{R}^d$, its input embedding is computed by

$$\boldsymbol{h}_i^V = \boldsymbol{h}_i^{\text{TE},V} + \boldsymbol{E}_{i,:}^{\text{PE}}, \tag{5}$$

for bond token $s_j^E$ with token embedding $\boldsymbol{h}_j^{\text{TE},E} \in \mathbb{R}^d$ and links atom $s_u^V$ and $s_v^V$, its input embedding is generated using

$$\boldsymbol{h}_j^E = \boldsymbol{h}_j^{\text{TE},E} + \boldsymbol{E}_{u,:}^{\text{PE}} + \boldsymbol{E}_{v,:}^{\text{PE}}, \tag{6}$$

as we assign no position embeddings to the special token $[\texttt{V}]$, its input embedding $\boldsymbol{h}_0$ equals its token embedding. The positional embeddings we inject into the input representations are flexible and able to encode complete structural information. Finally, the input embeddings can be formulated as

$$\boldsymbol{H}_0^{\text{SMILES}} = [\boldsymbol{h}_0, \boldsymbol{h}_1^V, \ldots, \boldsymbol{h}_n^V, \boldsymbol{h}_1^E, \ldots, \boldsymbol{h}_m^E]^\top \in \mathbb{R}^{l_{\text{SMILES}} \times d}, l_{\text{SMILES}} = n + m + 1. \tag{7}$$

From here we assume for $j = 1, \ldots, n$, $\boldsymbol{x}_j^V$ in $\boldsymbol{H}_0^{\text{Graph}}$ and $\boldsymbol{h}_j^V$ in $\boldsymbol{H}_0^{\text{SMILES}}$ represent the same atom $j$ in $M$, with such assumption for every $1 \leq r, s \leq n$, attention interaction from the $r$-th token to $s$-th token stands for the same atom-atom correlation in SMILES Transformer and Graph Transformer.

### 3.3    SMILES TRANSFORMER WITH KNOWLEDGE DISTILLATION

In this section we discuss the knowledge distillation from graph Transformer (GT) to SMILES Transformer (ST), a knowledge transfer process that bridges models with distinct input representations.

**Challenges**    For distillation between Transformer layers in language models, features like intermediate states and self-attention distributions can be used as transferred knowledge (Sun et al., 2019b; Jiao et al., 2019; Wang et al., 2020). However, we can not bring most of existing methods to distillation from GT to ST due to unmatched token sequences, for an input token sequence of ST has $l_{\text{SMILES}} = n + m + 1$ tokens, while for GT it has $l_{\text{Graph}} = n + 1$ tokens. For instance, if we simply distill the GT atom token embeddings or transfer the atom-atom attention weights, experiments have shown this unbalanced teaching (no bond token representations or atom-bond token interactions in ST are supervised) only results in subtle improvements.

**Global View of the Distillation Method**    Illustrated by Figure 2, our proposed method for distilling knowledge from a GT layer to a ST layer has two parts. First, both models use a virtual token for updating molecular embeddings, so we perform a straightforward feature distillation on the virtual token embeddings to transfer the teacher's learned molecular feature while avoiding the unmatching issue when distilling on other tokens. Second, to transfer the GT attention weights without disrupting the attention interactions in ST, we introduce a supportive attention bias module to ST layers. This attention bias module learns the atom-atom attention distributions in the teacher model, and highlights the learned pairwise interactions by adding scalar biases in the calculation of attention weights. We describe the detailed distillation scheme in the following paragraphs.

**Molecular Feature Distillation**   Consider the distillation from layer $p$ of GT to layer $q$ of ST with corresponding output molecule emebddings $\boldsymbol{x}_0^p \in \mathbb{R}^{d_V}$ and $\boldsymbol{h}_0^q \in \mathbb{R}^d$. Then the feature distillation loss can be defined as

$$\mathcal{L}_{\text{feat}}^{(p,q)} = \text{MSE}(\boldsymbol{x}_0^p, \boldsymbol{h}_0^q \boldsymbol{W}_{\text{feat}}), \tag{8}$$

here MSE() specifies mean square loss function, $\boldsymbol{W}_{\text{feat}} \in \mathbb{R}^{d \times d_V}$ is a learnable linear transformation matrix to transform features of the student into the same space as the teacher's features, which can also be removed if $d = d_V$.

**Biased Multi-head Attention**   We have introduced the multi-head attention mechanism (MHA) in Section 3.1. Here define biased Transformer layer by adding learnable attention biases to MHA, which serve as a side module to learn and emphasize knowledge from teacher attention weights. Following the notations, for every attention head $i$, the attention bias matrix is calculated by

$$\boldsymbol{B}_i = (\boldsymbol{H}\boldsymbol{W}_i^{Q,\text{bias}})(\boldsymbol{H}\boldsymbol{W}_i^{K,\text{bias}})^\top, \tag{9}$$

where $\boldsymbol{W}_i^{Q,\text{bias}}, \boldsymbol{W}_i^{K,\text{bias}} \in \mathbb{R}^{d \times d_k}$ are learnable attention bias projections not shared among heads. With an associated attention bias mask $\boldsymbol{M} \in \{0,1\}^{l \times l}$, the computing step of biased multi-head attention block can be defined as

$$\boldsymbol{A}_i = \text{Softmax}(\frac{\boldsymbol{Q}_i \boldsymbol{K}_i^\top}{\sqrt{d_k}} + \boldsymbol{M} \odot \boldsymbol{B}_i), \text{head}_i = \boldsymbol{A}_i \boldsymbol{V}_i, \text{for } i = 1, \ldots, h, \tag{10}$$

$$\text{Biased-MHA}(\boldsymbol{H}, \boldsymbol{M}) = \text{Concat}(\text{head}_1, \ldots, \text{head}_h)\boldsymbol{W}^O. \tag{11}$$

**Attention Weight Distillation via Biases**   The proposed ST-KD consists of standard Transformer layers and biased Transformer layers. We perform the distillation from layer $p$ of GT to biased Transformer layer $q$ of ST-KD. Suppose layer $p$ outputs the attention weight tensor $\mathbf{A}^p \in [0,1]^{h' \times l_{\text{Graph}} \times l_{\text{Graph}}}$, where $h'$ is the number of attetion heads and $l_{\text{Graph}} = n + 1$. Our goal is to transfer knowledge in $\mathbf{A}^p$ to layer $q$ via attention biases. We first stack bias matrices of each attention head in MHA of layer $q$ into the attention bias tensor

$$\mathbf{B}^q = [\boldsymbol{B}_1, \boldsymbol{B}_2, \ldots, \boldsymbol{B}_h] \in \mathbb{R}^{h \times l_{\text{SMILES}} \times l_{\text{SMILES}}}, \tag{12}$$

and we introduce a attention mask matrix $\boldsymbol{M} \in \{0,1\}^{l_{\text{SMILES}} \times l_{\text{SMILES}}}$ to mask out pairwise attention interactions except atom-atom or atom-vn (virtual node) ones. Specifically, the definition is

$$M_{i,j} = \begin{cases} 1, \text{ if } 1 \le i \le n \text{ and } 0 \le j \le n, \\ 0, \text{ else.} \end{cases} \tag{13}$$

Notably, $\boldsymbol{M}$ filters out attention interactions that won't match $\mathbf{A}^p$, and also serves as attention bias mask in the following attention calculations. Then we multiply $\mathbf{B}^q$ by $\boldsymbol{M}$, and reduce its size to fit $\mathbf{A}^p$ (removed elements are all zeros, according to definition of $\boldsymbol{M}$) in

$$\tilde{\mathbf{B}}^q = (\boldsymbol{M} \odot \mathbf{B}^q)_{:,0:l_{\text{Graph}},0:l_{\text{Graph}}} \in \mathbb{R}^{h \times l_{\text{Graph}} \times l_{\text{Graph}}}, \tag{14}$$

where $\boldsymbol{M}$ is broadcasted at the element-wise product. Finally, by applying the softmax function on the last dimension of $\tilde{\mathbf{B}}^q$, a weight distribution that matches $\mathbf{A}^p$ is predicted from the attention biases. By penalizing the distance between the computed distribution and masked $\mathbf{A}^p$, the attention bias module is able to learn the knowledge inside teacher attention weights. We formulate the attention weight distillation loss as

$$\mathcal{L}_{\text{attn}}^{(p,q)} = \text{MSE}(\boldsymbol{M}_{0:l_{\text{Graph}},0:l_{\text{Graph}}} \odot \mathbf{A}^p, \text{Softmax}(\boldsymbol{W}_{\text{attn}} \tilde{\mathbf{B}}^q)), \tag{15}$$

here $\boldsymbol{W}_{\text{attn}} \in \mathbb{R}^{h' \times h}$ is an optional learnable linear projection used when the teacher and student have different numbers of attention heads. Unexpectedly, in experiments we find the simple mean square loss function is the best choice for model convergence, rather than cross entropy or KL-divergence.

**Final Loss Function**   A task-related loss $\mathcal{L}_{\text{task}}$ is also need during supervised training. Using the above objectives, we can unify the loss function of ST-KD to

$$\mathcal{L} = \mathcal{L}_{\text{task}} + \alpha \sum_{(p,q) \in P_{\text{feat}}} \mathcal{L}_{\text{feat}}^{(p,q)} + \beta \sum_{(p,q) \in P_{\text{attn}}} \mathcal{L}_{\text{attn}}^{(p,q)}, \tag{16}$$

where $P_{\text{feat}}$ and $P_{\text{attn}}$ are the set of all GT-ST layer pairs for feature and attention weight distillation, $\alpha$ and $\beta$ stands for the weight for feature distillation loss and attention weight distillation loss.

## 4 EXPERIMENTS

We first conduct knowledge distillation experiments on the recent PCQM4M-LSC dataset (Hu et al., 2021) for quantum chemistry regression, which is the latest large-scale standard dataset for molecular property prediction and contains more than 3.8M molecules. Then we investigate the transfer learning capacity of ST-KD on several molecular property prediction tasks, with models pretrained by knowledge distillation in the previous step. Finally, we perform tests to support our claim on the efficiency superioiriy of SMILES-based models over graph-based models, and run ablation studies to further inspect each component of ST-KD. A detailed description of datasets can be found in Appendix A.1.

### 4.1 KNOWLEDGE DISTILLATION ON PCQM4M-LSC

**Baselines** For graph-based methods, we compare ST-KD with GCN (Kipf & Welling, 2016) and GIN (Xu et al., 2018) with their variants with virtual node attached to improve graph property prediction performance (Gilmer et al., 2017). They are standard baselines and achieve good performances on official leaderboard [1]. In terms of SMILES-based models, we test the MLP models over the ECFP fingerpint which is derived using a variant of Morgan algorithm (Morgan, 1965), and a SMILES Transformer ST-BASE that uses a standard SMILES tokenizer and sinusoidal positional encodings. We also apply data augmentation on ST-BASE by adding 5 random equivalent SMILES for every data in the training set.

**ST-KD Settings** We pick Graphormer (Ying et al., 2021) as the teacher model since Graphormer is built upon the Transformer architecture and achieves current state-of-the-art performance on the PCQM4M dataset. We have reproduced the training process of Graphormer on PCQM4M dataset with official code[2] and reported our results[3]. Graphormer has 12 Transformer layers with hidden dimension 768 and 32 attention heads.

We build the SMILES Transformer with 3 Transformer layers followed by 3 biased Transformer layers. We set the hidden dimension $d$ to $512$, number of attention heads $h$ to $16$, dimension of FFN layers to $2048$. For distillation layer mappings, we perform feature distillation and attention weight distillation simultaneously from the last three layers of Graphormer to the three biased Transformer layers, i.e. the set of all GT-ST distillation layer pairs is $P_{\text{feat}} = P_{\text{attn}} = \{(9,3), (10,4), (11,5)\}$. For loss weights, we set $\alpha = 0.5$ and $\beta = 2.0$. All models are trained on 1 or 2 NVIDIA RTX 3090 GPU for up to 2 days. A complete description of training details is available in Appendix A.2.1.

| | Model | #Params | Train | Validation |
|---|---|---|---|---|
| Graph-based | GCN | 2.0M | 0.1522 | 0.1702(0.1684*) |
| | GIN | 3.8M | 0.1379 | 0.1533(0.1510*) |
| | GCN-VIRTUAL | 4.9M | 0.1317 | 0.1542(0.1536*) |
| | GIN-VIRTUAL | 6.7M | 0.1206 | 0.1421(0.1396*) |
| | Graphormer | 47.1M | 0.0590 (0.0582*) | **0.1274** (0.1234*) |
| SMILES-based | MLP-ECFP | 15.8M | 0.1032 | 0.2113 |
| | ST-BASE | 19.1M | 0.1483 | 0.2031 |
| | ST-KD | 20.4M | 0.0877 | **0.1379** |

Table 1: Results on PCQM4M-LSC measured by MAE. * indicates results cited from OGB-LSC paper (Hu et al., 2021) and Graphormer paper (Ying et al., 2021).

**Results** Table 1 summarizes model performances on PCQM4M-LSC dataset. ST-KD surpasses all previous SMILES-based models by a very large margin and performs better than the best GNN baseline GIN-VIRTUAL, which proves that the performance gap between SMILES-based models

---

[1]`https://github.com/snap-stanford/ogb/tree/master/examples/lsc/pcqm4m#performance`

[2]`https://github.com/microsoft/Graphormer`

[3]Training is performed on 2 NVIDIA RTX 3090 GPUs for 2 days with batchsize 1024.

and graph-based models can be overcome with the proposed strategies. Moreover, performance of SMILES Transformer attains a magnificent boost from 0.2062 to 0.1379, augmented with knowledge distillation and SMILES preprocess techniques we propose. Ablation studies on it can be found in Section 4.4. Still, ST-KD has a long way to go before catching up with its Graphormer teacher, but the progress we make so far has shown the extensive potentials of SMILES-based models.

## 4.2 MOLECULAR PROPERTY PREDICTION

In this section, we evaluate performances of ST-KD on popular molecular property prediction datasets QM9, QM8, QM7, FreeSolv and HIV in MoleculeNet (Wu et al., 2018). We mainly explore the transfer learning capacity of ST-KD by fine-tuning the model trained by knowledge distillation on PCQM4M-LSC in the previous step.

**Baselines** In this section, we benchmark ST-KD with GCN-VIRTUAL, GIN-VIRTUAL, Graphormer, MLP-ECFP and ST-BASE mentioned above in Section 4.1, along with MoleculeNet (best performances collected in (Wu et al., 2018)), MoLFormer-XL (Ross et al., 2021) and AttentiveFP (Xiong et al., 2019), which is a graph-based deep model that generates molecular fingerprints with local and global attentive layers. For Graphormer, we initialize its weights with the final checkpoint from PCQM4M-LSC training in Section 4.1.

**Settings** For dataset split, we split the dataset randomly into 8:1:1 as training, validation and test sets if no predefined split is available. For all models, we train them for 3 times with different random seeds, and report the means and standard deviations of performances. For ST-KD, in all datasets, we use the same model hyperparameters as section 4.1 and model weights are initialized with the checkpoint of best validation set performance saved in distillation on PCQM4M-LSC. Appendix A.2.2 presents detailed training procedures. Results are reported in Table 2.

| Dataset
Metric | QM9
Multi-MAE↓ | QM8
Avg-MAE↓ | QM7
MAE↓ | FreeSolv
RMSE↓ | HIV
ROC-AUC↑ |
|---|---|---|---|---|---|
| GCN-VIRTUAL | 1.3682±0.0482 | 0.0123±0.0004 | 74.081±2.182 | 1.013±0.069 | 75.99±1.0 |
| GIN-VIRTUAL | 1.2248±0.0554 | 0.0111±0.0004 | 69.682±1.718 | 0.852±0.075 | 77.07±0.8 |
| MoleculeNet | 2.350* | 0.0150±0.0020* | 94.7±2.7* | 1.150* | 79.20* |
| AttentiveFP | 1.292 | 0.0130±0.0006 | 66.2±2.713 | 0.962±0.197 | 79.10±1.2 |
| Graphormer | 0.9168±0.0301 | 0.0091±0.0002 | 44.342±1.419 | 0.860±0.082 | 80.51* |
| MLP-ECFP | 1.2021±0.0573 | 0.0129±0.0004 | 65.845±1.321 | 0.995±0.076 | 74.15±1.1 |
| MoLFormer-XL | 0.6804* | 0.0102* | - | **0.2308*** | **82.2*** |
| ST-BASE | 1.1862±0.0510 | 0.0169±0.0006 | 52.348±3.211 | 1.071±0.089 | 72.10±1.4 |
| ST-KD | **0.6215±0.0117** | **0.0080±0.0002** | **43.373±1.770** | 0.895±0.065 | 80.32±0.7 |

Table 2: Results on molecular property prediction datasets. * indicates results cited from (Wu et al., 2018; Ross et al., 2021; Ying et al., 2021), ↑ for higher is better, ↓ contrarily. Additional information on ST-KD's performance on QM9 can be found in Appendix A.2.3.

**Results** ST-KD shows outstanding performance on the QM datasets, being state-of-the-art and outperforms both Graphormer and pretrained MoLFormer-XL. On FreeSolv and HIV, MolFormer-XL shows leading performance due to large-scale pretraining, while ST-KD still delivers competitive results with performable graph-based baselines. Compared to ST-BASE, ST-KD gains significant performance improvement on all downstream molecular property prediction tasks, which confirms the effectiveness of our proposed distillation method. Despite being outperformed by the Graphormer teacher on PCQM4M-LSC, on downstream tasks, ST-KD performs better on QM datasets and slightly worse on FreeSolv and HIV, indicating that SMILES-based models can fit better on certain tasks than graph-based models. Compared to MoLFormer-XL (Ross et al., 2021), which requires pretraining on 16 NVIDIA V100 GPUs for 208 hours with at least 81M parameter size [4], ST-KD is light-weight (20.4M, 32-bit float), efficient at training (2 NVIDIA RTX 3090 GPUs

---

[4]No MolFormer-XL code is available, we estimate this given that MoLFormer-XL has 12 layers with hidden dimension 768, and we assume parameters are stored with 32-bit float numbers.

for 80 hours, including training Graphormer), and achieves better results on QM8 and QM9. This demonstrates that knowledge distillation is an effective and efficient way to raise the performance of SMILES-based models to state-of-the-art level, with contrast to large-scale pretraining.

## 4.3 EFFICIENCY TESTS

**Settings**   We follow the OGB-LSC official guide for measuring total SMILES-to-target inference time in [5], and apply all related parameters. To replicate a real-world scenario, we run tests on PCQM4M-LSC and a ZINC subset containing 1M molecules sampled from the public ZINC15 database (Sterling & Irwin, 2015), which represents for real molecular data in drug discovery. All models take raw SMILES strings as input, with their inference time being evaluated on a single NVIDIA GeForce RTX 3090 GPU and an Intel(R) Xeon(R) Platinum 8260C CPU @ 2.30GHz, 256G RAM with warm up. All models are directly taken from Section 4.1. The inference time is measured by millisecond (ms) per molecule. We report the results by the mean of 3 runs in Table 3.

| Dataset | | PCQM4M-LSC | | ZINC Subset | |
| Input | | Graph | SMILES | Graph | SMILES |
|---|---|---|---|---|---|
| Graph-based | GCN-VIRTUAL | 0.121 | 0.627 | 0.204 | 0.906 |
| | GIN-VIRTUAL | 0.118 | 0.627 | 0.209 | 0.913 |
| | Graphormer | 0.399 | 2.795 | 0.573 | 4.035 |
| SMILES-based | MLP-ECFP | - | 0.352 | - | 0.467 |
| | ST-BASE | - | 0.149 | - | 0.230 |
| | ST-KD | - | 0.192 | - | 0.271 |

Table 3: Results of inference time tests, measured by ms per molecule.

**Results**   Table 3 summarizes the inference time of all models. It can be observed that ST-KD reaches a 3-14× higher inference speed from raw SMILES strings than graph-based models, which proves our statements. Compared to ST-BASE, ST-KD runs relatively slower with the additional SMILES preprocessing step and attention bias computations, but has significantly better performance. We can infer from the Table that the SMILES-to-Graph conversion is a serious efficiency bottleneck for graph-based models, as we have mentioned in Section 1.

**Comparison with SMILES Pretrain Models**   We are unable to run inference time tests of most SMILES pretrain models since code is not publicly available, but we can make reliable conjectures about their inference speed with model hyperparameters. Considering MoLFormer-XL, which has 12 attention layers with hidden dimension 768, we can reasonably expect it to run at least 2-3 times slower than ST-KD (6 layers with hidden dimension 512) at inference. This also confirms the efficiency superiority of ST-KD over SMILES pretrain models.

## 4.4 ABLATION STUDIES

We run a series of ablation studies to further look into the importance of each component in the designed ST-KD architecture on the knowledge distillation experiments with PCQM4M-LSC dataset. Other model hyperparameter settings and training procedures stay the same as in Section 4.1. Table 4 demonstrates that all techniques we have plugged into ST-KD are necessary to raise the perfomance of SMILES Transformer to its best level. With the SMILES pre-transformation, the Transformer gains increased performance, since the input sequences now consist of well-structured tokens and meaningful input emebddings. As for knowledge distillation, both type of the distilled knowledge contribute to the teaching process. The performance of ST-KD can be optimized only with the combination of feature distillation and attention weight distillation.

---

[5]In **Measuring the Test Inference Time** of `https://github.com/snap-stanford/ogb/tree/master/examples/lsc/pcqm4m#performance` and code `test_inference_gnn.py` in this directory.

| SMILES Pre-transformation | Distillation | | Validation Perf. |
| | Feature Transfer | Attention Transfer | MAE |
|---|---|---|---|
| - | - | - | 0.2062 |
| ✓ | - | - | 0.1808 |
| ✓ | ✓ | - | 0.1672 |
| ✓ | - | ✓ | 0.1589 |
| ✓ | ✓ | ✓ | 0.1379 |

Table 4: Ablation studies of ST-KD on PCQM4M-LSC dataset with different components removed.

## 5 CONCLUSION

We have proposed ST-KD, a novel architecture for SMILES Transformer empowered by knowledge distillation and shown its competitive results on benchmarks. As high-throughput and high-performance molecular feature extractors, SMILES-based models exhibit significant practical potentials in handling large-scale molecular data. With initial results being encouraging, challenges still exist. Our distillation approach cannot be applied without a well-learned graph model as teacher. In distillation experiments, Graphormer outperforms its student well, so results of ST-KD can be further improved with a more effective knowledge transfer strategy. We leave them as future works.

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

# A    APPENDIX

## A.1    DATASETS

Statistics of datasets we use in this paper are shown in Table 5. PCQM4M-LSC is a large-scale graph property prediction dataset in the recent OGB-LSC Large-Scale Challenge (Hu et al., 2021), originally curated under the PubChemQC project (Nakata & Shimazaki, 2017). The task of PCQM4M-LSC is to predict HOMO-LUMO gap of molecules calculated by DFT (Density Functional Theory) with 2D structure of mulecules. Test labels of PCQM4M-LSC is now unavailable, and we use validation set to evaluate model performance. QM9, QM8 and QM7 are datasets containing organic molecules with up to 9, 8 and 7 heavy atoms (any atom that is not hydrogen) and their quantitative quantum-chemical properties. For QM7, we use a sanitized version by DeepChem (Ramsundar et al., 2019) available in `https://deepchemdata.s3-us-west-1.amazonaws.com/datasets/qm7.csv`. FreeSolv contains molecules with their corresponding quantitative hydration free energy. HIV contains molecules labeled by their HIV active labels, with a predefined scaffold data split.

| Datasets | PCQM4M-LSC | QM9 | QM8 | QM7 | FreeSolv | HIV |
|----------|------------|-----|-----|-----|----------|-----|
| **#molecules** | 3803453 | 133885 | 21786 | 6834 | 642 | 41127 |
| **#tasks** | 1 | 12 | 16 | 1 | 1 | 1 |
| **Task Type** | Regression | | | | | Classification |
| **Metric** | MAE | Multi-MAE | | MAE | RMSE | ROC-AUC |

Table 5: Statistics of Datasets.

## A.2    EXPERIMENT DETAILS

### A.2.1    KNOWLEDGE DISTILLATION ON PCQM4M-LSC

All parameters involved are listed in Table 6. During experiments we find by adding an initial stage when there is no task-related supervision, the model can have a faster convergence. Parameters of this initial stage is also shown.

### A.2.2    MOLECULAR PROPERTY PREDICTION EXPERIMENTS

For ST-KD, we load the model checkpoint with best performance on validation set saved during distillation as initial weights for molecular property prediction experiments in this section. In experiments we discover that results can benefit from changing the model dropout value at this finetune training process. All related parameters are listed in Table 7 and 8.

### A.2.3    ADDITIONAL INFORMATION ON PERFORMANCES OF ST-KD ON QM9 DATASET

QM9 is a dataset with 12 regression targets. We follow the conventions to standardize the targets and fit them simultaneously. Here we report ST-KD's test set performance on 12 tasks seperately.

| Parameter | Value |
|---|---|
| #Transformer Layers | 3 |
| #Biased Transformer Layers | 3 |
| Hidden Dimension $d$ | 512 |
| Key Dimension of Attention Block $d_k$ | 32 |
| Feed-Forward Network Dimension | 2048 |
| #Attention Heads $h$ | 16 |
| Dropout Value | 0.1 |
| Feature Distillation GT-ST Layer Pairs | $\{(9,3), (10,4), (11,5)\}$ |
| Attention Distillation GT-ST Layer Pairs | $\{(9,3), (10,4), (11,5)\}$ |
| Task Loss Weight | 1.0 |
| Feature Distillation Loss Weight $\alpha$ | 0.5 |
| Attention Distillation Loss Weight $\beta$ | 2.0 |
| Initial Stage Epochs | 5 |
| Initial Stage Task Loss Weight | 0.0 |
| Initial Stage Feature Distillation Loss Weight $\alpha$ | 1.0 |
| Initial Stage Attention Distillation Loss Weight $\beta$ | 4.0 |
| Optimizer | AdamW |
| Max Epochs | 200 |
| Batch Size | 256 |
| Learning Rate | 1e-4 |
| Weight Decay | 1e-5 |
| $\epsilon$ of AdamW | 1e-8 |
| $(\beta_1, \beta_2)$ of AdamW | (0.9,0.999) |

Table 6: Parameter settings of knowledge distillation experiment on PCQM4M-LSC.

| Parameter | Value |
|---|---|
| Optimizer | AdamW |
| Max Epochs | 100 |
| Batch Size | 80 |
| Dropout Value | 0.0 |
| Learning Rate | 1e-4 |
| Weight Decay | 1e-5 |
| $\epsilon$ of AdamW | 1e-8 |
| $(\beta_1, \beta_2)$ of AdamW | (0.9,0.999) |

Table 7: Parameter settings of molecular property prediction experiments on QM9, QM8 and QM7.

| Parameter | Value |
|---|---|
| Optimizer | AdamW |
| Max Epochs | 100 |
| Batch Size | 50 |
| Dropout Value | 0.1 |
| Learning Rate | 1e-4 |
| Weight Decay | 1e-5 |
| $\epsilon$ of AdamW | 1e-8 |
| $(\beta_1, \beta_2)$ of AdamW | (0.9,0.999) |

Table 8: Parameter settings of molecular property prediction experiments on FreeSolv and HIV.

| Task | MAE (Standardized) | MAE |
|---|---|---|
| mu | 0.2319±6.421e-3 | 0.3548±9.827e-3 |
| alpha | 0.03130±1.208e-3 | 0.2563±9.893e-3 |
| homo | 0.1082±8.654e-4 | 2.390e-3±1.912e-5 |
| lumo | 0.05173±7.760e-4 | 2.426e-3±3.640e-5 |
| gap | 0.06803±1.167e-3 | 3.232e-3±5.544e-5 |
| r2 | 0.06301±1.061e-3 | 15.73±0.2651 |
| zpve | 0.01357±1.634e-3 | 4.518e-4±5.440e-5 |
| u0 | 6.300e-3±2.160e-4 | 0.2534±8.654e-3 |
| u298 | 6.233e-3±1.886e-4 | 0.2497±7.554e-3 |
| h298 | 6.402e-3±5.657e-4 | 0.2564±2.266e-2 |
| g298 | 6.567e-3±2.494e-4 | 0.2631±9.993e-3 |
| cv | 0.02820±7.118e-4 | 0.1146±2.892e-3 |

Table 9: Performances of ST-KD on QM9, reported by seperate tasks.

