# OpenReview forum: "Stepping Back to SMILES Transformers for Fast Molecular Representation Inference"
_ICLR.cc/2022/Conference — ICLR 2022 Submitted_

### Official Review · Reviewer_Maqb · 2021-11-01

**Correctness:** 3
**Technical Novelty And Significance:** 3
**Empirical Novelty And Significance:** 2
**Recommendation:** 5
**Confidence:** 3

**Main Review:**

I am uncertain about the efficiency claim in this paper. According to my experience, SMILES-to-graph preprocessing is not that time-consuming (in my case probably 10 mins for 10M conversion in 1 process), and this is a preprocessing step, meaning that it only executes once for the future repeated use. Furthermore, a lot of molecule libraries are stored in .sdf format that we only need to read without conversion. I thus feel the motivation is exaggerated while I am also glad to listen to other reviewers.

Besides, it sacrifices the training efficiency to trade the inference one. According to Table 1, compared with GNNs, ST-KD is 3-10 times larger (so what is the training cost which is not reported?) considering training is usually much more expensive than inference. Since graph transformer is also used in training, SMILES-to-graph conversion exists thus the comparison would be training complexity.

I am also concerned with the tokenization step in section 3.2 that authors convert SMILES into a sequence composed of atom characters and bond characters. For instance, cc=cc will be converted to cccc= if I understand correctly. If it is the case, there will be two types of mistakes: 1) different molecules will produce the same token sequences as both cc=cc and c=ccc will be cccc=; 2) the same molecules will produce different token sequences as c=occ and ccc=o will be cocc= and ccco=, respectively. I do not judge on the second mistake since this is essentially the drawback of SMILES representations which is not permutation invariant, but I am worried about the first one.

**Summary Of The Paper:**

The paper targets speeding up the inference process (specifically, SMILES-to-graph preprocessing) in virtual screening by adopting SMILES-based transformer for molecule representation learning. To ensure the performance of SMILES-based transformer, authors propose to use graph-based transformer as teacher network to conduct knowledge distillation (ST-KD). Numerical results claim for the competitive performance and faster inference time.

**Summary Of The Review:**

I am concerned about the over-claim of efficiency and the correctness of tokenization.

---

> ### Author Response · Authors · 2021-11-09
> **Official Reply to Reviewer Maqb**
>
> Thanks for your constructive review. For your major concerns, here is our reply:
>
> 1. Uncertainties about the efficiency claims.
>
>    (a) We would believe our efficiency claims are well supported by section 4.3, and your experiences can also corroborate our test results. It can be inferred from table 3 that a single SMILES-to-graph conversion takes about 0.5ms on PCQM4M-LSC (results of GCN and GIN), while for your example, 10 mins for 10M conversion means it takes 0.6ms for a single conversion, which corresponds to our numbers.
>
>    (b) While it seems that SMILES-to-graph conversion is not time-consuming in medium-sized datasets considering these numbers, in circumstances when we need to extract molecular feature from a large-scale unexplored SMILES database, the efficiency superiority of SMILES-based methods can make a large difference, and this is what our model is designed for. For example, in PCQM4M we have tested, GCN takes 40 mins to extract all molecular features, Graphormer takes 180 mins, and ST-KD uses only 12 mins. Accordingly, we believe the efficiency of ST-KD is not overclaimed.
>
>    (c) As .sdf is a common data format for describing molecules (especially with 3D structure) in many chemical libraries, SMILES is still a universal descriptor for 2D molecular structure and is used by real-world chemical datasets like ZINC. Also, when distributing or matching 2D molecular structure, readable and ascii encoded SMILES strings are far more convenient than the .sdf files. We are convinced that fast SMILES models are indispensable when handling tasks on existing vast SMILES data.
>
> 2. Training efficiency issues.
>
>    Indeed, ST-KD has relatively higher training complexity compared to GNNs. But our main focus is the test inference efficiency, which means that after training process of ST-KD is completed, it can then serve as an accurate molecular feature extractor on future SMILES data, being substantially faster than GNNs. Furthermore, the distillation strategy we apply is more efficient and effective compared to SMILES pretrain models. For instance, **the new baseline MoLFormer-XL requires large-scale pertaining on ZINC+PubChem data, but ST-KD could surpass MoLFormer-XL on 2 tasks out of 4 (Table 2) with much less #parameters (20M vs 80M) and training costs (160 vs 3328 GPU hours).**
>
> 3. Questions on the tokenization step.
>
>    We believe there is a misunderstanding here. Aside from token embeddings, we also introduce positional embeddings to indicate the bond connectivity. For cc=cc, it will be converted to cccc= and the = character has positional embedding $E_{2,:}^{\text{PE}}+E_{3,:}^{\text{PE}}$, while for c=ccc, it will still be converted to cccc= but the = character has different positional embedding $E_{1,:}^{\text{PE}}+E_{2,:}^{\text{PE}}$ (Simplified symbols, please refer to section 3.2 for more details). Thus cc=cc and c=ccc are distinguishable in input representations, so mistake 1 does not exist. We have realized that the method descriptions in section 3.2 are not easy to read, and we will try to update them in the future.

---

> > ### Comment · Reviewer_Maqb · 2021-11-29
> > **Response**
> >
> > Thank you for your clarification. My concern 3 is addressed but 1&2 are still held. I keep my initial score.

---

> > > ### Author Response · Authors · 2021-12-01
> > > **Official Reply 2 to Reviewer Maqb**
> > >
> > > Thanks for your comments. We would still like to emphasize some facts corresponding to your first two concerns:
> > >
> > > 1. Are SMILES-to-graph conversion and featurization not a bottleneck? They surely are. Experiments show that **SMILES-to-graph conversion takes 5 times longer than inference in GCNs.**
> > >
> > > 2. Can preprocessing help? In some cases but not all. Largest drug datasets like ZINC contain over 230M molecules. Preprocessing for all of them not only takes long, but also costs unacceptable spaces to store the results. Indeed, sdf files save some time in retrieving the bonds, yet SMILES is still the most universal storage solution in most databases.
> > >
> > > 3. Is KD really that slow? The total training GPU hours of ST-KD is 160, vs 72 on GCN. We believe the extra training time is acceptable regarding to a 3-14$\times$ faster inference speed. Also, compared with other pre-train SMILES-based models, ST-KD has much less training cost, as we mentioned in the replies above.

---

### Official Review · Reviewer_xrsP · 2021-11-02

**Correctness:** 4
**Technical Novelty And Significance:** 3
**Empirical Novelty And Significance:** 4
**Recommendation:** 8
**Confidence:** 4

**Main Review:**

Strengths:
1.	The core ideas of this paper are interesting and well-implemented. To my knowledge, end-to-end SMILES-based molecular fingerprint generators are discussed by many prior works, but none of them have proposed to improve the model with knowledge from existing state-of-the-art graph models. Followed by the key insight, the authors have proposed effective methods to distill knowledge from graph Transformers to SMILES Transformers.
2.	Convincing experimental results. Results on PCQM4M and ablation studies have corroborated the effectiveness of distillation methods. The model also achieves outstanding performances on downstream tasks like QM9, where the 0.62 multi-mae surpasses all existing models I know by a large margin, including those with molecular conformation provided.
3.	Clear writing and illustrations. The authors have stated their motivations and implement details clearly with well-drawn figures. I find no overstatements in this paper.

Weaknesses:
1.	More experimental datasets would be welcome. The authors have performed molecular property prediction experiments on QM datasets and FreeSolv. More experiments would be welcome on more datasets, such as in MoleculeNet and OGB, to further evaluate the performances of ST-KD on different downstream molecular tasks.
2.	More experiments on parameter analysis would be good. The authors run the knowledge distillation experiments on PCQM4M with a fixed network setting. Additional experiments can be done to explore the effects of different network structure and training hyperparameters.


**Summary Of The Paper:**

High-throughput molecular representation for large chemical databases is one of the kernel requirements for many downstream applications, such as drug discovery and virtual screening etc. Based on the consideration that SMILES strings are the most common storage format for molecules, this paper provides a SMILES-based molecular representation learning model, named ST-KD, by using deep graph models to extract molecular feature from raw SMILES data without SMILES-to-graph conversion. Specifically, the proposed model adopts an end-to-end SMILES Transformer framework for molecular representation learning boosted by Knowledge Distillation. Experimental study shows that ST-KD shows competitive results on latest standard molecular datasets PCQM4M-LSC and QM9, with 3-14 inference speed compared with existing graph models.

**Summary Of The Review:**

This paper is based on an interesting idea of building a fast and high-quality SMILES-based molecular fingerprint generator with knowledge distillation from state-of-the-art graph-based models. The authors have described the proposed methods clearly, with their claims being supported by experimental results. Although more experiments can be performed to give a thorough evaluation to the model’s performance, I believe the paper should be recommended due to its novel idea and convincing results.

---

> ### Author Response · Authors · 2021-11-17
> **Official Reply to Reviewer xrsP**
>
> Thanks for recommending our paper! We have uploaded a revised version of our paper, and here is our detailed reply:
>
> 1. More datasets are needed.
>
>    In response to your advices, we have added experimental results on the HIV dataset, a molecular classification task in MoleculeNet. ST-KD can now get strong performance on HIV, similar to Graphormer. Results on HIV have provided another evidence of the strong generalization ability of ST-KD.
>
> 2. Experiments on network settings should be provided.
>
>    We are sorry for not conducting network setting experiments since we do not have enough 	computational resources. The current version of ST-KD has competitive performance with a small size of parameters and resonable training cost compared with SMILES pretrain models, so we would consider that the current network setting is enough to demonstrate the effectiveness of our method.

---

> > ### Comment · Reviewer_xrsP · 2021-12-01
> > **thanks to the response**
> >
> > i like this work and my comments has been addressed and thus i keep my score

---

### Official Review · Reviewer_PfwA · 2021-11-02

**Correctness:** 4
**Technical Novelty And Significance:** 3
**Empirical Novelty And Significance:** 3
**Recommendation:** 3
**Confidence:** 4

**Main Review:**

The knowledge distillation framework is quite interesting, where the author carefully provide the way to perform it Fig2 and Sec 3.2.

My major concern about this work is the performance of the SMILES-based transformer is not worse than the graph-based model. In fact, in many cases SMILES-based transformers outperform graph-based models (or at least comparable performances). Why is there a need to perform knowledge distillation to learn a SMILES Transformer (student) from graph Transformer (teacher) ?


The SMILES-based model (e.g. ST-BASE) used as baseline in this paper seem not a competitive SMILES-based model. Other competitive SMILES based models should be included in the comparison. For example, https://github.com/pschwllr/MolecularTransformer
which has a different way of tokenizing and utilizing SMILES-based input. The SMILES-Transformer's performance can be largely boosted by simple augmentation trick, e,g using random-smiles as augmentation when training the model. The performance of this model may outperform graph-based models. Can you please include the comparison against this work?


Additionally, if we refer to table 2, it seems that there is not a clear improvement of graph-based models over ST-BASE.

Can the author provide intuition of why the proposed methods ST-KD can outperform both graph and SMILES based methods in table 2, when it is trained to minimize the difference between graph transformer and SMILES transformer (by eq 16)? Should the teacher ( the graph-based methods)  be the upper bond?



**Summary Of The Paper:**

The authors provide a knowledge distillation framework to transformer knowledge from graph transformer (teacher) to SMILES transformer (student), where the  SMILES transformer (student) is forced to mimic graph transformer (teacher) 's desired property.

**Summary Of The Review:**

The authors provide an interesting knowledge distillation framework to learn molecule embedding, however, the reviewer is not fully onboard with the motivation of doing the knowledge distillation (e.g. learning SMILES based transformer from graph-based transformer), since smiles-based transformer has a competitive performance.

---

> ### Author Response · Authors · 2021-11-17
> **Official Reply to Reviewer PfwA**
>
> We greatly appreciate your insightful review. In the newly refined version of our paper, we have updated experiments according to your questions and advices. We truly hope our new revisions can help to solve your concerns.
>
> 1. SMILES-based transformer is not worse than the graph-based model, so the need to perform knowledge distillation to learn a SMILES Transformer (student) from graph Transformer (teacher) is not clear.
>
>    In the latest version, **we added Graphormer results on downstream tasks**. We can easily observe from Table 2 that **Graphormer perfroms consistently better than other graph-based baselines and SMILES-based models without knowledge distillation or pretraining**.  And the effect of knowledge distillation is manifested in the significant performance improvements achieved from ST-BASE (uncomparable against Graphormer) to ST-KD (better or slightly worse compared to Graphormer) on PCQM4M-LSC and all downstream datasets.
>
> 2. SMILES-based baselines are not strong enough. Models like MolecularTransformer should be included in the comparison.
>
>    Thanks for your technical advices. MolecularTransformer is built for chemical reaction prediction, so we cannot directly run its code for comparison. Instead, in the latest version, we updated ST-BASE with tokenizer and data augmentation tricks in MolecularTransformer model (https://github.com/pschwllr/MolecularTransformer), and its results can now reflect MolecularTransformer's capability on molecular property prediction tasks. However, we only witnessed a subtle performance improvement of the updated ST-BASE over the original one. We assume the tokenization step and augmentation tricks do not have substantial impact on the model's expressive power.
>
>    In addition to the updated ST-BASE, we also introduced a strong SMILES-based baseline, MoLFormer-XL, a very recent SMILES pretrain model which has shown nearly state-of-the-art performances on many tasks. Yet ST-KD can surpass MoLFormer-XL on 2 tasks out of 4 (Table 2), with much less training cost (160 vs 3328 GPU hours) and parameters (20M vs 80M). This demonstrates the competitive performance and outstanding efficiency of ST-KD.
>
> 3. The reasons for why ST-KD can outperform both graph and SMILES based methods are not clear, and we do not know if the graph-based teachers should be the upper bound. Intuitions should be provided.
>
>    This question is insightful, and we would explain it from the following perspectives:
>
>    (a) The experiments have shown that the teacher is not the upper bound in downstream tasks for ST-KD, and it's not uncommon in knowledge distillation research when the student outperforms the teacher. For example, in https://arxiv.org/pdf/1909.10351.pdf, the student model TinyBERT outperforms its BERT teacher in many downstream GLUE tasks.
>
>    (b) Our knowledge distillation method is built to improve the performance of SMILES-based models with structured knowledge learned by graph Transformers. This means that in our case, the teacher and student do not have much difference in model capacity (not a large teacher to a small student in model compression), and they do not take in the same form of data. Therefore, though a trained-from-scratch SMILES model performs bad, **we can not assume the distilled student to perform consistently worse than the teacher in all kinds of tasks since their potential expressive power can not be precisely ranked.** From our perspective, the competitive performance of ST-KD on downstream molecular tasks have shown the effectiveness of our distillation method (ST-BASE can not do that), and SMILES-based models can be better suited for certain tasks than graph-based models (like quantum-mechanical regression tasks).

---

### Official Review · Reviewer_2H4F · 2021-11-08

**Correctness:** 2
**Technical Novelty And Significance:** 3
**Empirical Novelty And Significance:** Not applicable
**Recommendation:** 5
**Confidence:** 4

**Main Review:**

Strengths:
(1) Knowledge distillation between graph transformer and SMILES transformer is proposed for molecular representation learning.
(2) Feature distillation and attention transfer losses are used for knowledge transfer.
(3) ST-KD achieves strong performance, similar to graph-based models, on molecular property prediction tasks, with a fast inference time.

Weaknesses:
(1)	Recent SMILES transformer models trained on large data have shown performance similar to state-of-the-art graph-based models without any knowledge distillation, see e.g. Ross et al 2021 https://arxiv.org/abs/2106.09553. Such works should be discussed in comparison to ST-KD.
(2)	ST-KD performs slightly worse than Graphformer on PCQM4M-LSC as in Table 1. At the same time, the comparisons of ST-KD with Graphformer on QM* and Freesolv are not presented in Table 2. Therefore, it is not evident to what extent knowledge transfer is successful on QM* and Freesolv.
(3)	 It is not clear to what extent ST-KD performance is generalizable beyond the tasks discussed in the paper – for example, Moleculenet classification tasks or on bigger molecules.
(4)	Graph-to-text knowledge distillation has been investigated before, e.g. https://aclanthology.org/2020.emnlp-main.551.pdf. Such works should be at least cited in this work.
(5) Inference time comparison is not shown for datasets other than PCQM4M-LS.


**Summary Of The Paper:**

Authors propose knowledge distillation from graph Transformer (GT) to SMILES Transformer toward learning a molecular representation with similar performance to Graph-based models while having low inference time (by avoiding smiles-to-graph conversion bottleneck).

Main contributions are:

(1)  An end- to-end SMILES Transformer for molecular representation learning is proposed that is boosted by graph transformer to smiles transformer knowledge distillation.
(2) Biased transformer layers are designed to achieve knowledge transfer; learnable SMILES token embedding are used to encode structural information.
(3) ST-KD shows competitive results on PCQM4M-LSC and QM9, with fast inference compared to existing graph models.
(4)

**Summary Of The Review:**

The work is interesting, as it explores the idea of knowledge distillation from a Graph transformer to a SMILES transformer for performant molecular prediction with fast inference. However, the idea of cross-modal knowledge distillation is not novel. While it is well understood that the SMILES transformer after KD provides the advantage of fast inference, it is not clear if knowledge distillation is the way to get to SOTA performance with fast inference, as recent pre-trained SMILES-based efficient transformer models are providing competitive performance.  Also, the work does not provide any strong evidence of the  generalizability of the proposed approach.

---

> ### Author Response · Authors · 2021-11-17
> **Official Reply to Reviewer 2H4F**
>
> We appreciate your valuable comments and suggestions. In the latest version of our paper, we have added some experiments and explanations. Hopefully they may resolve some of your concerns.
>
> 1. Comparisons with SOTA SMILES Transformers.
>
>    We understand that a previous weakness of our paper was that we ignored some recent, strong SMILES Transformers on molecular benchmarks. Meanwhile, we found that most of these networks have massive amounts of parameters and depend on training on very large datasets. Specifically, we added the comparison between MoLFormer-XL and ST-KD in our updated version in Section 4.2 and 4.3. Indeed, MoLFormer-XL was a strong SMILES baseline which obtained SOTA performances on many tasks, yet we observed that **ST-KD could surpass MoLFormer-XL on 2 tasks out of 4 (Table 2) with much less #parameters (20M vs 80M) and training costs (160 vs 3328 GPU hours)**. We believe this comparison still corroborates the contribution of our paper.
>
> 2. ST-KD vs Graphormer on QM and Freesolv*
>
>    In our latest version, we **added the results of Graphormer on downstream tasks including QM and FreeSolv.** Significant improvements were consistently achieved between ST-BASE (uncomparable against Graphormer) and ST-KD (better or slightly worse compared to Graphormer) on PCQM4M-LSC and all downstream datasets. Specifically on QM9 and QM8, ST-KD exceeds MoLFormer with much less training cost. We believe the significant performance boost demonstrates the effect of the proposed knowledge distillation method.
>
> 3. Generalizing ST-KD on HIV tasks.
>
>    According to your suggestion, we added experimental results on the HIV dataset, a molecular classification task in MoleculeNet. Results on HIV shows that with knowledge distillation on large datasets, fine-tuned ST-KD can achieve results similar to Graphormer. As molecules in the HIV dataset generally have larger sizes, this provides an evidence of the generalization capability of ST-KD.
>
> 4. Discussions on graph-to-text knowledge distillation methods.
>
>    Thank you for the suggestions, and we added discussions on cross-modal knowledge distillation in Section 2. Indeed, SMILES are strings, and ST-KD can be seen as a special case of graph-to-text distillation, yet honestly speaking, our model was not directly motivated by these methods. Nevertheless, we would like to stress that ST-KD uses specifically designed approaches to handle SMILES and different, molecule-based methods on distillation.
>
> 5. Inference time comparisons on datasets other than PCQM4M-LS
>
>    To resolve your concern, we provided efficiency tests on a subset of ZINC15 database in the latest version. This new test can reflect a real-world drug discovery scenario, and ST-KD keeps its high inference efficiency against other methods.

---

### Author Response · Authors · 2021-11-17
**A new version of our paper has been updated.**

In this version, we have added experiments and comparisons with new baselines to address the reviewers' concerns. Major updates of the revised paper are listed below:

### Section 2

**We have added investigations into cross-modal knowledge distillation**.

### Section 4.1

**ST-BASE baseline has been updated with a smarter tokenizer and data augmentation tricks.** Now ST-BASE use a tokenizer that splits SMILES into chemical tokens, instead of single characters. Data augmentation is also applied on ST-BASE by adding 5 random equivalent SMILES for every data in the training set. **All results related to ST-BASE has been updated in section 4.1 and 4.2.**

### Section 4.2

**A new baseline MoLFormer-XL (https://arxiv.org/abs/2106.09553) has been added.** MoLFormer-XL is a pretrain SMILES model with competitive performance on downstream datasets, but also has large model size and high training costs. **Comparisons of MoLFormer-XL with ST-KD on performances, training costs and inference speed have been discussed in section 4.2 and 4.3.**

**Graphormer results have been added.** Graphormer now performs as the best graph-based baseline. **Note that to give a fair comparison, we initialize its weights with the final checkpoint from PCQM4M-LSC training in Section 4.1.**

**Results on a new dataset HIV have been added.** HIV is a MoleculeNet classfication dataset.

### Section 4.3

**Efficiency tests on a subset of ZINC15 database have been added.** This new test replicates a real-world drug discovery scenario, and ST-KD keeps its high performance.

We also want to make clarifications on problems frequently asked by reviewers:

### **ST-KD is not compared against strong SMILES baselines.**

In the newest revision, we have introduced the pretrain baseline MoLFormer-XL, and updated the ST-BASE model with training tricks.

MoLFormer-XL requires **large-scale pertaining (ZINC+PubChem data, 3328 V100 GPU hours)**, and ST-KD can **surpass MoLFormer-XL on 2 downstream tasks out of 4**, with **much less training cost (160 RTX 3090 GPU hours, including training the teacher)** and presumably much faster inference speed (Section 4.3). The comparison between ST-KD and MoLFormer-XL demonstrates the effectiveness and efficiency of knowledge distillation methods we propose. With this, we can train SMILES models that has competitive performance and fast inference speed with much less computational resources than pretraining.

ST-BASE is also updated, now it is equipped with many training tricks in prior related works, including a better tokenizer and data augmentation with random equivalent SMILES. The performance is not improved to a large extent, though.

### **Some SMILES models can surpass graph models on downstream tasks, this contradicts with previous claims.**

We have added Graphormer results on downstream tasks in this version of our paper, which now consistently performs as the best graph-based baseline. Now we can easily observe from Table 2 that without distillation or pretraining, SMILES-based models cannot outperform Graphormer at downstream tasks. This supports our previous claims.

---

### Decision · Program_Chairs · 2022-01-20

**Decision:**

Reject

**Comment:**

The paper  builds  fast and high-quality SMILES-based molecular embeddings  by distilling  state-of-the-art graph-based models teachers.
This has the advantage of speeding inference time w.rt to graph based methods.

The reviews were split regarding the motivation of the work, in the sense of why not train directly on SMILES instead of distilling graph based methods that are in some tasks behind SMILES transformer. Authors provided clarifications in the rebuttal showing that on Knowledge distillation of graph models  surpasses  SMILES only model training.

I think given the experimental nature of the paper the main motivation of the paper should be better clarified and supported with more experimentation and downstream tasks.